# Magnetostrictive Performance of Electrodeposited Tb_x_Dy_(1−x)_Fe_y_ Thin Film with Microcantilever Structures

**DOI:** 10.3390/mi11050523

**Published:** 2020-05-21

**Authors:** Hang Shim, Kei Sakamoto, Naoki Inomata, Masaya Toda, Nguyen Van Toan, Takahito Ono

**Affiliations:** 1Department of Mechanical Systems Engineering, Tohoku University, Sendai 980-8579, Japan; inomata@nme.mech.tohoku.ac.jp (N.I.); mtoda@nme.mech.tohoku.ac.jp (M.T.); nvtoan@nme.mech.tohoku.ac.jp (N.V.T.); 2Micro-Nanomachining Research Education Center, Tohoku University, Sendai 980-8579, Japan; kei.sakamoto.e1@tohoku.ac.jp; 3Micro System Integration Center (µSiC), Tohoku University, Sendai 980-8579, Japan

**Keywords:** magnetostriction, thin film, Terfenol-D, Tb_x_Dy_(1−x)_Fe_y_, electrodeposition

## Abstract

The microfabrication with a magnetostrictive Tb_x_Dy_(1−x)_Fe_y_ thin film for magnetic microactuators is developed, and the magnetic and magnetostrictive actuation performances of the deposited thin film are evaluated. The magnetostrictive thin film of Tb_x_Dy_(1−x)_Fe_y_ is deposited on a metal seed layer by electrodeposition using a potentiostat in an aqueous solution. Bi-material cantilever structures with the Tb_0.36_Dy_0.64_Fe_1.9_ thin-film are fabricated using microfabrication, and the magnetic actuation performances are evaluated under the application of a magnetic field. The actuators show large magnetostriction coefficients of approximately 1250 ppm at a magnetic field of 11000 Oe.

## 1. Introduction

Magnetostriction is a useful property of ferromagnetic materials that causes strain during the process of magnetization. The strain of the magnetostriction materials can be controlled by a magnetic field [1,2,3,4,5,6]. Also, the strain itself can generate a magnetic field, referred to as inverse-magnetostrictive effect or Villari effect [1,2,3]. Magnetostriction can be quantified by the magnetostrictive coefficient which can be positive or negative and is defined as the generated strain when a magnetic field causing magnetization saturation is applied. For example, Fe, Ni, and Co are well known as magnetic materials, which show small magnetostrictive coefficients of −14, −50, and −93, respectively [1,2,3]. It is known that Fe alloys containing a rare earth material exhibit a large magnetostrictive coefficient at room temperature, and those alloys are referred to as “giant magnetostriction material” [1,2,3,4]. Among those, Terfenol-D (Tb_0.3_Dy_0.7_Fe_2_) exhibits a large magnetostrictive coefficient up to 1400 ppm at a magnetic field of ~2 kOe [1,2,3,4,7]. In addition, Galfenol, Fe_0.8_Ga_0.2_, is known as a material, which shows a large magnetostriction up to 400 ppm [5,6] and CoFe shows 260 ppm as well [8,9,10,11,12]. Since these discoveries, those materials have emerged as a smart material for microdevices, including actuators [13,14], wireless sensors, biosensors [15,16], energy harvesting devices [17], and atomic force microscopy [18].

Most of the devices based on the giant magnetostriction materials are made from bulk materials. However, the importance of the thin-film technology for those materials have gained for realizing miniaturized smart actuators and devices. Many methods have been reported for thin-film preparation, including pulsed laser deposition [7], sputtering [10,11,19,20,21,22,23,24,25,26,27,28], and electrochemical deposition [5,6,9,29]. For the film deposition of Terfenol-D, the sputtering method has been reported because of the simple approach, high film uniformity, and low roughness. However, the substrate must be heated at a temperature higher than 400 °C for crystallization, because the sputter-deposited Terfenol-D films are amorphous state at low temperatures, which show a low magnetostrictive performance [23,24,25]. Those sputtered films exhibit a magnetostriction coefficient approximately 1/3 (540 ppm) of the bulk value without annealing and 2/3 (920 ppm) of the bulk value with annealing at 450 °C [19,20,21]. Electrochemical deposition has advantages for simplicity, low cost, and compatibility with batch fabrication, etc., and some of the researches have been reported regarding magnetostrictive thin films including Galfenol [5,6], CoFe [9], and TbFe_2_ [29].

Rare earth atoms including Tb and Dy are generally difficult to deposit by electrodeposition using an aqueous solution because those materials have reduction potential <−2 V (eg., Tb^+3^ + 3e^−^ = Tb: −2.28 V, Dy^3+^ + 3e^−^ = Dy: −2.6 V); therefore, hydrogen evolution makes the aqueous solution unstable [26]. Chemical additives add to reducing electrochemical deposition potential and improve film quality [29,30,31,32,33]. Gong et al. demonstrated the deposition of TbFe_2_ film in an aqueous solution with a rare earth metal complex [29].

In this work, Terfenol-D films are deposited by electroplating and the performances of the deposited films are evaluated using energy-dispersive X-ray spectroscopy (EDX) and vibrating-sample magnetometer (VSM) analysis. Microcantilever bi-material structures are fabricated, and the magnetostriction performances are evaluated.

## 2. Experimental

A Tb_x_Dy_(1−x)_Fe_y_ thin film was deposited by electrodeposition using a potentiostat with three conventional electrodes: working, counter, and reference electrodes. This aqueous electrolyte was prepared from rare earth sulfate salts and iron salts. In order to reduce the deposition potential, those ions are chelated by citric acid and tartaric acid. Thus, the electrolyte is formed by mixing deionized 150 mL water with following chemicals, i.e., Tb_2_(SO_4_)_3_ 0.3 g, Dy_2_(SO_4_)_3_ 0.7 g, FeCl_3_ 0.5 g, FeSO_4_ 1.8 g, tartaric acid 3 g, citric acid 0.5 g, KCl 33 g, NaOH 1.1 g [34]. Electrodeposition was proceeded on a 300 nm-thick Cr-Cu seed layer deposited on a silicon on insulator (SOI) wafer with a 100 nm-thick Si/300 nm-thick buried oxide layer/ 750 µm-thick Si handling layer. The electrodeposition was performed at 40 °C for 2.5 h at a working electrode potential of (−925)–(−950 mV). The typical deposition rate of the Tb_x_Dy_(1−x)_Fe_y_ film is approximately 100 nm/h. However, the deposition speed was varied by deposition area, the resistance of the seed layer, the distance between working and counter electrode, and temperature of the electrolyte. The compositions of the deposited films are analyzed by energy-dispersive X-ray spectroscopy (EDX). The magnetization properties for the in-plane direction of the film are measured using a vibrating sample magnetometer (VSM). For the analysis of surface morphology, atomic force microscopy (AFM) and magnetic force microscopy (MFM) are used. The magnetostriction coefficients are measured from the bi-material cantilever actuation using an optical microscope with a special resolution of 4.4 µm corresponding to one imaging pixel. The sample preparation for the Tb_x_Dy_(1−x)_Fe_y_ cantilevers is performed by microfabrication. The details of the fabrication process are described later.

Figure 1 shows the typical EDX result of a 300 nm-thick Tb_x_Dy_(1−x)_Fe_y_ thin film prepared at −930 mV of potential, which formed on the SOI wafer with the Cr-Cu seed layer. For the analysis, the substrate and seed layer components are ignored. Generally, the composition of electrodeposited alloys can be adjusted by the applied potential because of the reduction potential difference of each component. However, applicable potential has a limited window; for the potential *V* > −700 mV, Fe atoms do not deposit. For the case of the potential *V* < −1055 mV, hydrogen evolution happens, and the electrolyte is degraded. To maximize the magnetostriction performance, the atomic concentration ratio of rare earth and iron atoms must satisfy 1:2 [1,2,3,4]. Thus, the ideal weight percentage of Fe atoms is approximately 40%, and the atomic percentage is approximately 66%. The Fe concentration can be controlled by the working electrode potential, as shown in Figure 2, which shows approximately 1% to 2% of the atomic percentage change of Fe atoms with 1 mV of potential variation. The optimal electrochemical potential can be found at −930 mV for the film deposition on the Cu seed layer. The composition fractions of Tb, Dy, and Fe were analyzed to be approximately 12.6, 22 and 65.4 atomic %, respectively; thus, the film composition is approximated to be Tb_0.36_Dy_0.64_Fe_1.9_.

The in-plane magnetization analysis of the 200 nm-thick Tb_0.36_Dy_0.64_Fe_1.9_ sample with the SOI wafer was proceeded using VSM, as a result is shown in Figure 3. It is found that the coercive magnetic field is approximately 285 Oe. The magnetization is saturated at approximately 5000 Oe, and magnetization starts to decrease. Thus, the effective magnetic field for magnetostrictive actuation can be regarded to be in the range of 285–5000 Oe. It is reported that the bulk Terfenol-D shows 63 Oe of coercive magnetic field [27], 2000 Oe of saturation magnetic field and 1 T of saturation magnetization [1]. Compared with bulk value, the saturation magnetization of the electrodeposited film is 40% lower than that of the bulk value. There are several suspected reasons for this degradation. One is that the film composition ratio is slightly different from the ideal value of Terfenol-D. The different composition ratio shows different magnetization characteristics [1]. Another reason is the magneto crystalline anisotropy effect [35]. The magneto crystalline anisotropy plays important roles in magnetic domain rotation, magnetization and magnetostriction. An electrodeposited magnetostriction GaFe film shows uncontrolled lattice orientation as reported [6]. These random lattice structures affect low magnetization and magnetostriction characteristics [6,9]. The possible other reason for this is the oxide impurity of the film, which pins the magnetic domains [21], decreases saturation magnetizations and increases the coercive magnetic field [21,22,26].

Figure 4 and Figure 5 show the atomic force microscopy and magnetic force microscopy images of the deposited film with a thickness of 300 nm. In the atomic force microscopy image, the Tb_0.34_Dy_0.65_Fe_1.9_ film has a fine grain microstructure with a diameter of ~170 nm, which is larger than the grain size of reported sputtered films ~50–55 nm [19,20]. Generally, the grain size of electrodeposited films is much bigger than that of the as-deposited sputtered film. From the magnetic force microscopy images, the Tb_0.34_Dy_0.65_Fe_1.9_ film shows a large magnetic domain and grain boundary. This is one of the evidences that electrodeposited Tb_0.34_Dy_0.65_Fe_1.9_ film possesses a polycrystalline structure. Sputtered Terfenol-D films without annealing shows amorphous state crystallinity. The amorphous Terfenol-D film show maze shape magnetic domain image from magnetic force microscopy. Polycrystalline state Terfenol-D, however, shows magnetic domain structure sillier to crystal grain structure [21].

The magnetostriction coefficient is generally defined by generated strain under the application of a magnetic field. However, in the case of thin films, it is difficult to measure the strain of the films directly. Thus, an analytical model using the displacements of bi-material cantilevers is employed to evaluate the magnetostriction coefficients [9,36,37], in which the effective magnetostriction coefficient value $\lambda_{eff}$ can be calculated from the displacements of the bi-material cantilever structures with an application of magnetic fields in parallel and perpendicular against the longitudinal direction of the cantilever, as given by [37],
(1)$$\lambda_{eff}=\frac{2\left(D_{∥}-D_{⊥}\right)E_{s}t_{s}^{2}\left(1+v_{f}\right)}{9l^{2}E_{f}t_{f}\left(1+v_{s}\right)}$$
where $D_{∥}$ is the displacement in parallel to the magnetic field, $D_{⊥}$ is the displacement in perpendicular to the magnetic field, $E_{s}$ and $v_{s}$ are Young’s modulus and Poisson ratio of the substrate material, respectively, $E_{f}$ and $v_{f}$ are the Young’s modulus and Poisson ratio of the magnetostrictive material, $t_{f}$ and $t_{s}$ are the film thicknesses of the magnetostrictive layer and the substrate, respectively, and l is the length of the cantilever. The spring constant *k_cantilever_* of the cantilever structure, and the force *F* generated by the magnetostrictive film can be approximated as given by following equations [38],
(2)$$k_{cantilever}=4+6*\frac{t_{f}}{t_{s}}+4*{\left(\frac{t_{f}}{t_{s}}\right)}^{2}+\frac{E_{f}}{E_{s}}*{\left(\frac{t_{f}}{t_{s}}\right)}^{3}+\frac{E_{f}}{E_{s}}*\frac{t_{s}}{t_{f}},$$
(3)$$F=k_{cantilever}D$$
where *D* is the displacement of the cantilever. The elastic properties of thin-films and microstructures are generally almost same with that of the bulk [39]; thus, for this calculation, the Young’s modulus of each layer is supposed to be the bulk value of silicon and Terfenol-D, i.e., $179\times 10^{9}\;Pa$ and $50\times 10^{9}\;Pa$, respectively, and the Poisson ratios of the silicon substrate and the Tb_0.34_Dy_0.65_Fe_1.9_ film are supposed to be 0.22 and 0.3, respectively [28,35].

Table 1 shows the typical dimensions of the fabricated cantilever. From Table 1 and Equation (2), the effective spring constant of the composite cantilever is calculated to be 26.4 N/m.

The fabrication process of the bi-material cantilever is shown in Figure 6. The Tb_0.34_Dy_0.65_Fe_1.9_ (Terfenol-D) film is deposited on the SOI wafer with a Cu seed layer by electrodeposition and patterned by ion beam milling with a photoresist mask. To prevent the oxidation of the Tb_0.36_Dy_0.64_Fe_1.9_ film, a 25 nm-thick Si_3_N_4_ thin film is deposited on the magnetostrictive film by sputtering. After etching the handling Si layer from the backside, the cantilever structures are released by etching the buried oxide using vapor HF etching. The SEM images of the fabricated bi-material microcantilever structures are shown in Figure 7. Owing to the stress of the films, the cantilevers are slightly bent upward.

## 3. Result and Discussion

Using an electromagnet, a magnetic field of 0–11 kOe is applied to the fabricated cantilevers along the parallel direction of the cantilever. The actuation is observed by a microscope, as the typical result is shown in Figure 8, where the magnetic field was applied along horizontal direction in parallel to the cantilever length direction. The magnetostrictive film on the Si cantilever has 91 MPa tensile stress as observed from the initial bending. Theoretically, Terfenol-D is known as positive magnetostriction material. The Terfenol-D material will extend toward the magnetic field direction. When a magnetic field is applied to the cantilever, the cantilever will be bent downward because of the magnetostriction effect.

With an optical microscope, this displacement could be observed, as shown in Figure 8. An application of the magnetic field actuates the cantilever downward with displacement $D_{∥}$. Figure 9 shows the observed displacements as a function of applied magnetic field for three cantilevers with different lengths, and Figure 10 shows the magnetostriction coefficients calculated using Equation (1). Figure 11 shows the generated forces of each cantilever calculated from the cantilever deflection under various magnetic fields using Equation (3). The maximum force can be estimated to be approximately 65 mN.

The magnetostriction coefficient $\lambda_{eff}$ can be calculated using Equation (1), as shown in Figure 10, where *D*_⊥_is supposed to be negligible. The actuation is saturated at approximately 5000 Oe. This actuation characteristic seems to be reasonable with the VSM result. At 11000 Oe, the Tb_0.36_Dy_0.64_Fe_1.9_ film shows a magnetostrictive coefficient of approximately 1250 ppm in strain, which is comparable to 1400 ppm of the magnetostriction coefficient of the bulk Terfenol-D [1,2,7]. This is the highest value among reported magnetostriction coefficients of the Tb_x_Dy_(1−x)_Fe_y_ films [19,20,21,22,23,24,25,26,27,28]. Also, from the displacement data and deflection data, the energy density of the thin film actuator can be calculated from the stored elastic energy $W_{el}$ in the cantilever [37]. The stored elastic energy is given by
(4)$$W_{el}=\frac{E_{s}}{1-v_{s}}*{\left(\frac{1}{R}\right)}^{2}z^{2}Lldz\;$$
where R is the radius of the curvature of the cantilever, L is the length of the cantilever, l is the width of the cantilever and $E_{s}$, $v_{s}$ are the Young’s modulus and Poisson ratio of the cantilever, respectively. Specific dimensions and parameters used are shown in Table 1. The radius *R* of the curvature *R* and deflection *D* of the cantilever is given by
(5)$$R^{-1}=-\frac{\beta }{\beta^{2}-\beta +\frac{1}{3}}\sigma_{int}\frac{h_{f}}{h_{s}^{2}}\frac{1-v_{s}}{E_{s}}=-6\sigma_{int}\frac{h_{f}}{h_{s}^{2}}\frac{1-v_{s}}{E_{s}}$$
(6)$$D=-3\sigma_{int}\frac{h_{f}}{h_{s}^{2}}\frac{1-v_{s}}{E_{s}}L^{2}$$
where $\sigma_{int}$ is the initial stress of the cantilever, and β is the constant of the neutral plane of the cantilever. The Young’s modulus of silicon is approximately three times larger than that of Terfenol-D, also the thickness of the Tb_0.34_Dy_0.65_Fe_1.9_ film is very thin in comparison with the Si layer; thus β = 1/2 is the proper assumption in this model. As a consequence, the stored elastic energy in the cantilever is given by
(7)$$W_{el}=\overset{\beta h_{s}}{\underset{\left(\beta -1\right)h_{s}}{\int }}\frac{Es}{1-vs}{\left(\frac{1}{R}\right)}^{2}z^{2}Lldz=\frac{E_{s}}{1-v_{s}}{\left(\frac{1}{R}\right)}^{2}z^{2}Llh_{s}^{3}\left(\beta^{2}-\beta +\frac{1}{3}\right)$$

The energy density *E*_density_ of the magnetostrictive film is obtained by dividing the stored elastic energy by the volume *V*_f_ of the magnetostrictive film, as given by
*E*_density_ = *W_el_*/*V_f_*(8)

In the actual calculation, the maximum energy density is calculated from the actuated cantilever deflection using Equation (8). The calculated and reported energy densities of the film and bulk [1,6] are summarized in Table 2. The variation of the calculated energy density seems to be large for three cantilevers, it may come from the cantilever dimension errors (possibly ±50 µm) caused by alignment error and side etching in microfabrication.

Figure 12 shows the comparison of the energy density of the Tb_0.36_Dy_0.64_Fe_1.9_ film with another actuator. It is found that this magnetostrictive film can produce very high energy density for actuation. It is found that the energy density can be higher than that of piezoelectric material (PZT) that is widely used for microelectromechcanical devices.

In Table 3, the magnetostriction coefficient of this Tb_0.36_Dy_0.64_Fe_1.9_ film is compared with that of bulk materials and reported magnetostrictive thin films. The electrodeposited Galfenol and CoFe films show a relatively low performance in comparison with the sputtered films. It is considered that this low performance possibly comes from random lattice orientation [6,9]. The sputtered and annealed Terfenol-D and CoFe show better than non-annealed film [11,19,20]. This better performance comes from improved crystallinity and grain size. It can be concluded that the electrodeposited Tb_0.36_Dy_0.64_Fe_1.9_ film shows excellent magnetostriction performance than that of other types of magnetostrictive thin films and has high potential ability for the application to microelectromechanical systems (MEMS) including magnetic actuators, energy harvesters, and microsensors.

## 4. Conclusion

This paper reported the performance of a Tb_0.36_Dy_0.64_Fe_1.9_ film deposited by electrodeposition at 40 °C. The deposited Tb_0.36_Dy_0.64_Fe_1.9_ film shows the coercive magnetic field 285 Oe and the saturation magnetic field 5000 Oe. From AFM and MFM analysis, the film has ~170 nm grain size. At 11 kOe magnetic field, the Tb_0.36_Dy_0.64_Fe_1.9_ film shows approximately 1250 ppm of magnetostriction coefficient. Moreover, the energy density of the film is calculated to be 100,000~165,000 J/m^3^. These performances are almost the same to those of bulk Terfenol-D. As a consequence, the electrodeposited Tb_0.36_Dy_0.64_Fe_1.9_ film has a high potential ability for magnetic actuator, energy harvesting, and sensor applications.

## Figures and Tables

**Figure 1 micromachines-11-00523-f001:**
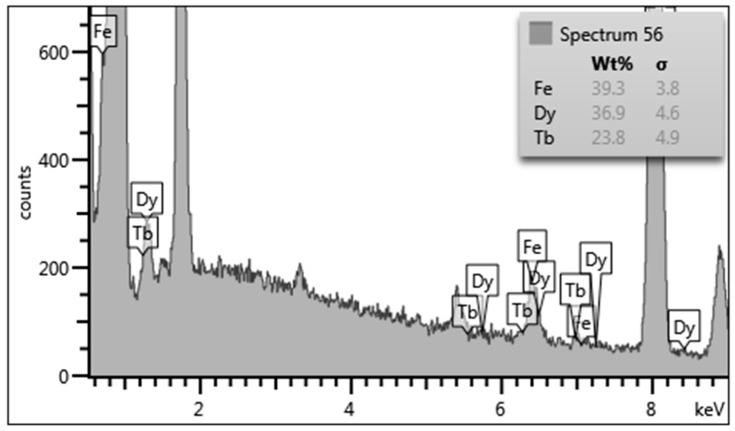
Energy-dispersive X-ray spectroscopy ***(***EDX) spectrum of the Tb_x_Dy_(1−x)_Fe_y_ film formed at an electrochemical potential of −930 mV.

**Figure 2 micromachines-11-00523-f002:**
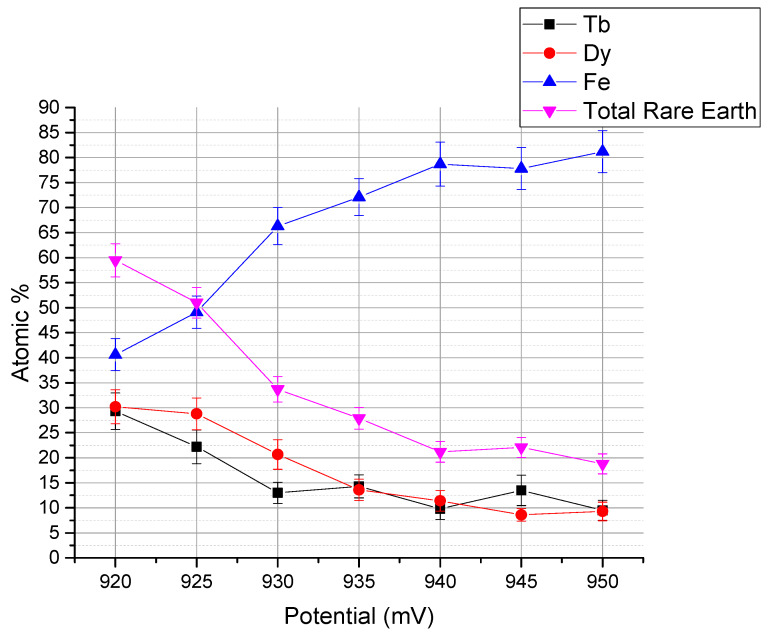
Composition dependence on the applied working electrode potential on a Cu seed layer.

**Figure 3 micromachines-11-00523-f003:**
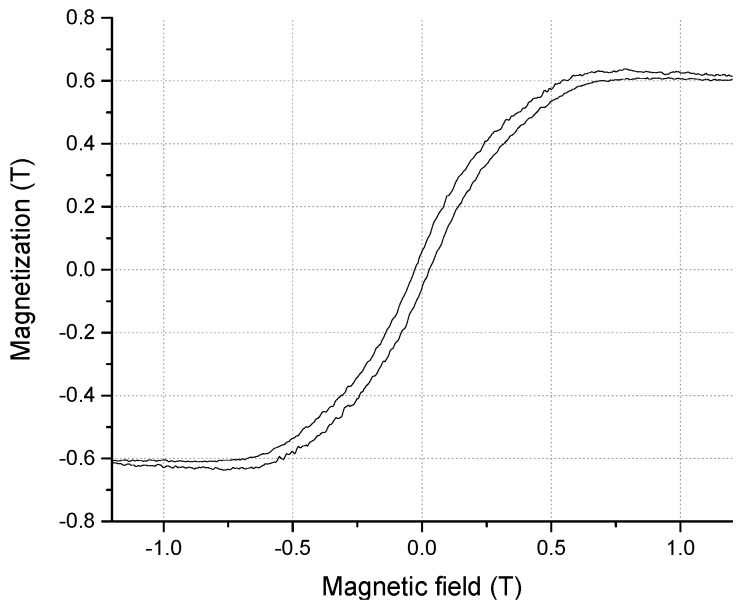
In-plane magnetization measurement of the Tb_x_Dy_(1−x)_Fe_y_ film using vibrating sample magnetometer (VSM).

**Figure 4 micromachines-11-00523-f004:**
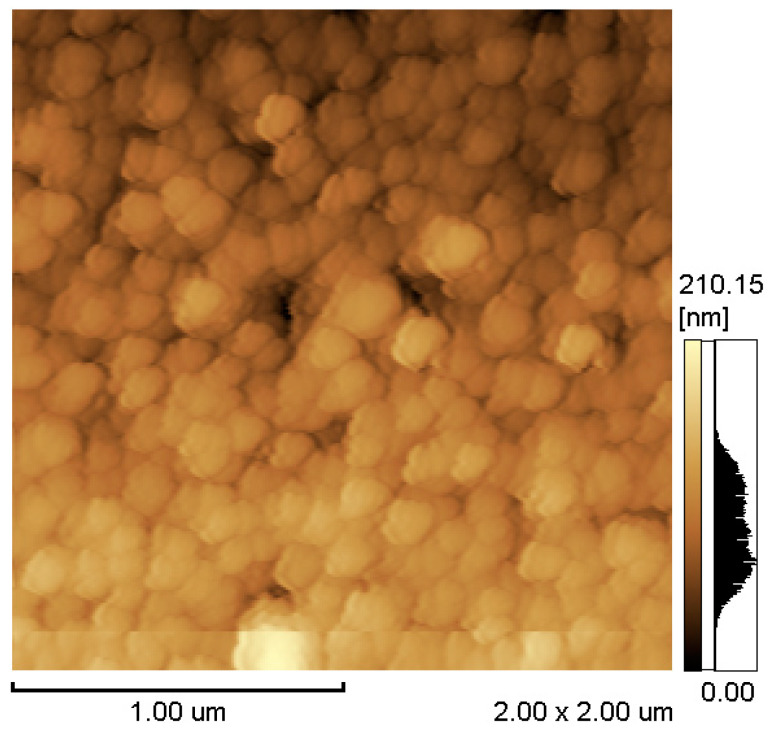
Atomic force microscopy image of the deposited film.

**Figure 5 micromachines-11-00523-f005:**
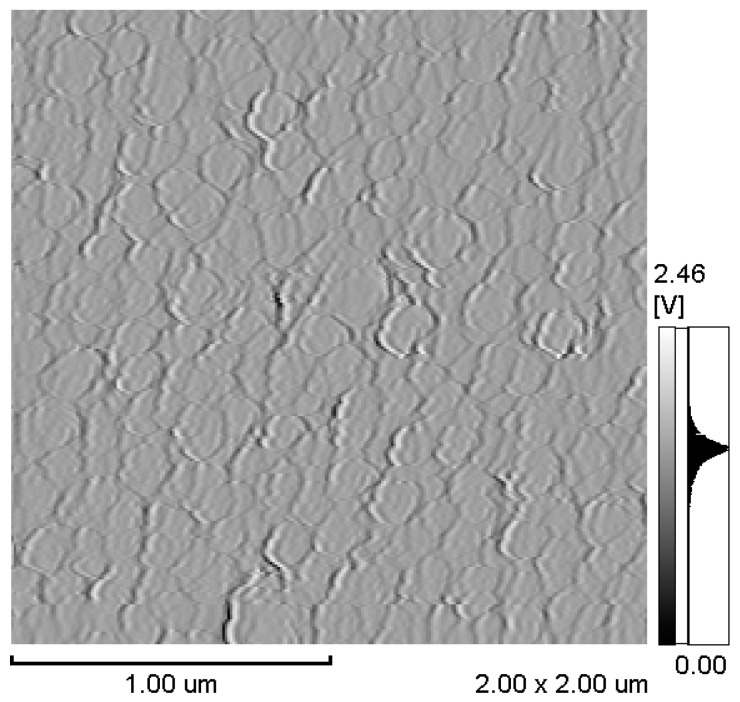
Magnetic force microscopy the deposited film.

**Figure 6 micromachines-11-00523-f006:**
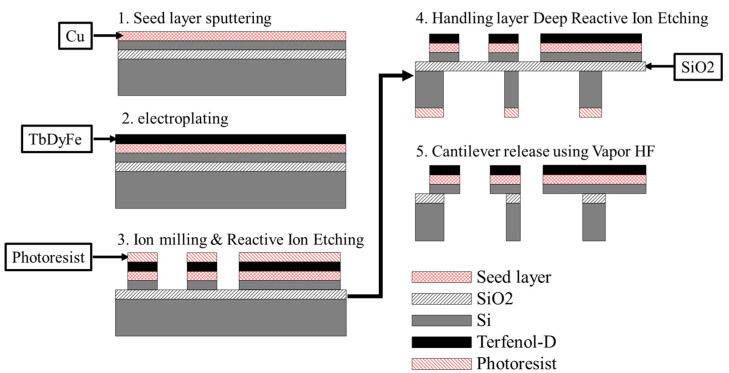
Fabrication process of the magnetostrictive bi-material cantilevers.

**Figure 7 micromachines-11-00523-f007:**
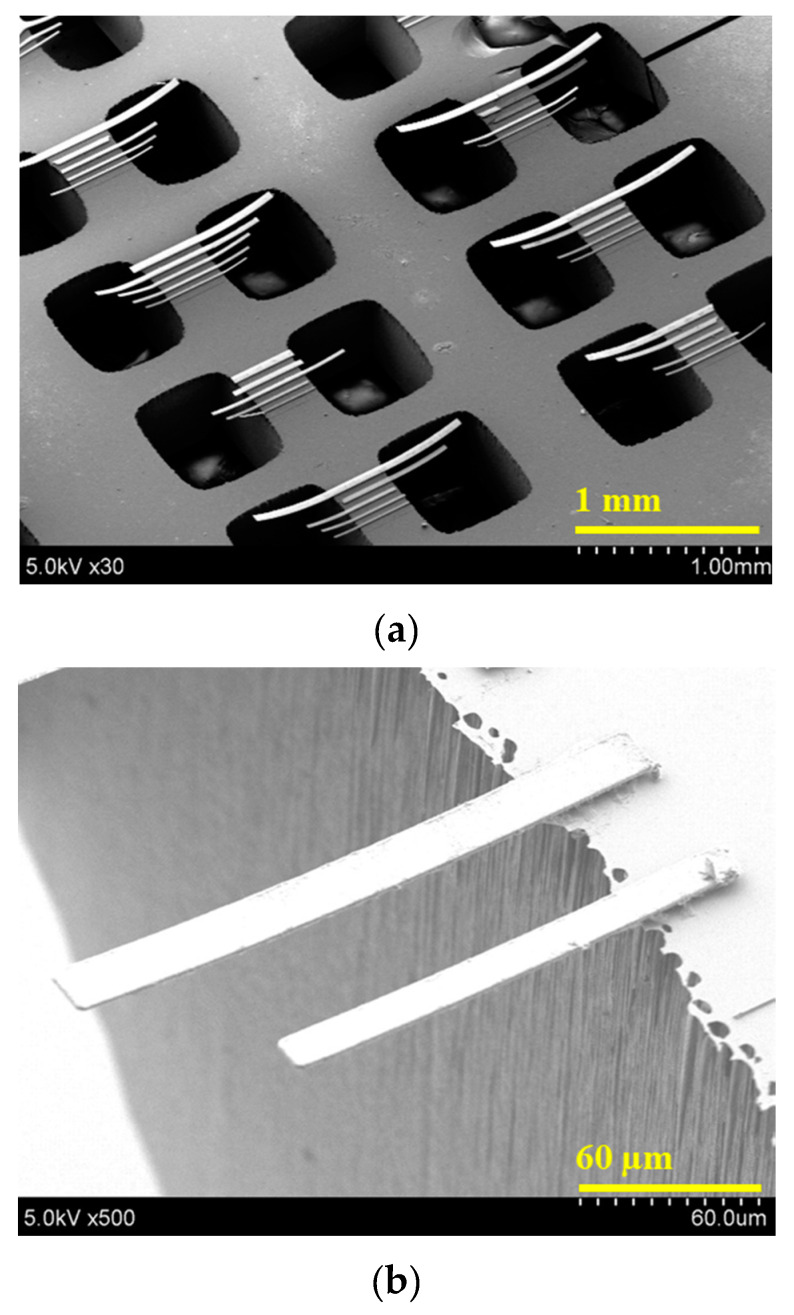
SEM images of fabricated Tb_0.34_Dy_0.65_Fe_1.9_ bi-material cantilevers. (**a**) Low magnification of bi-material cantilevers; (**b**) High magnification of bi-material cantilevers.

**Figure 8 micromachines-11-00523-f008:**
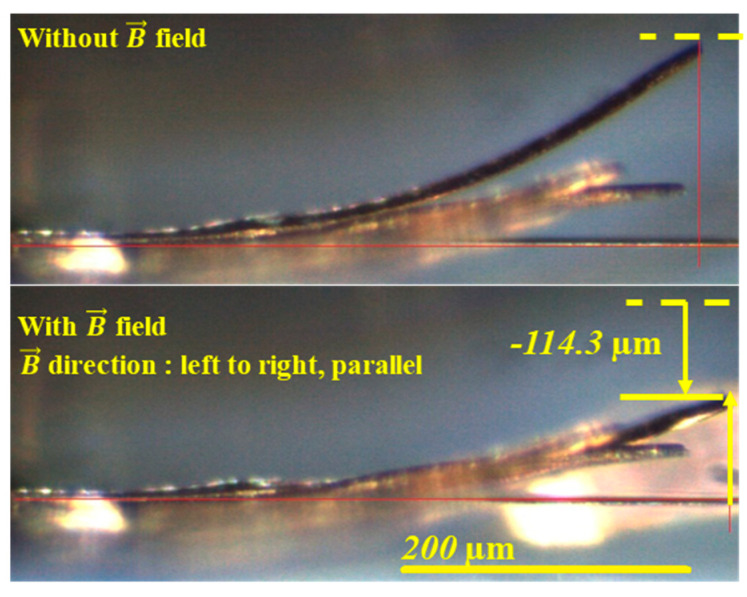
Optical images of side view for the typical magnetostrictive actuation of the Si- Tb_0.34_Dy_0.65_Fe_1.9_ bi-material cantilever for the cases without magnetic field and with a magnetic field of 11 kOe along the cantilever direction.

**Figure 9 micromachines-11-00523-f009:**
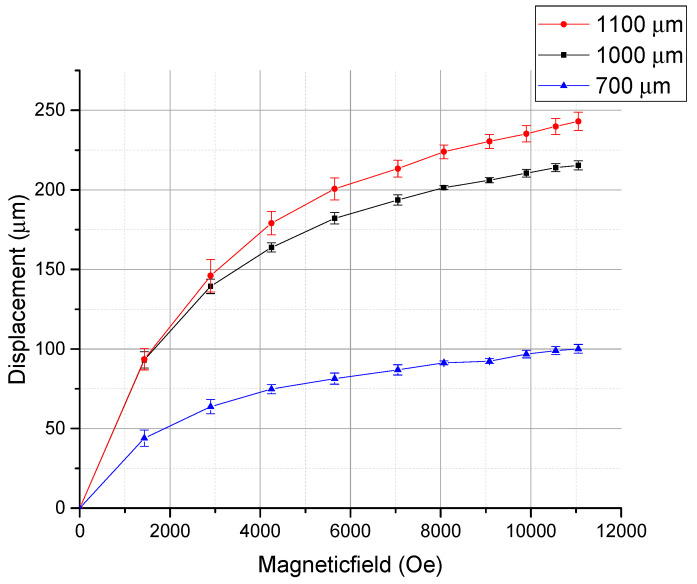
Observed displacements of the cantilevers with different lengths (700, 1000, 1100 µm).

**Figure 10 micromachines-11-00523-f010:**
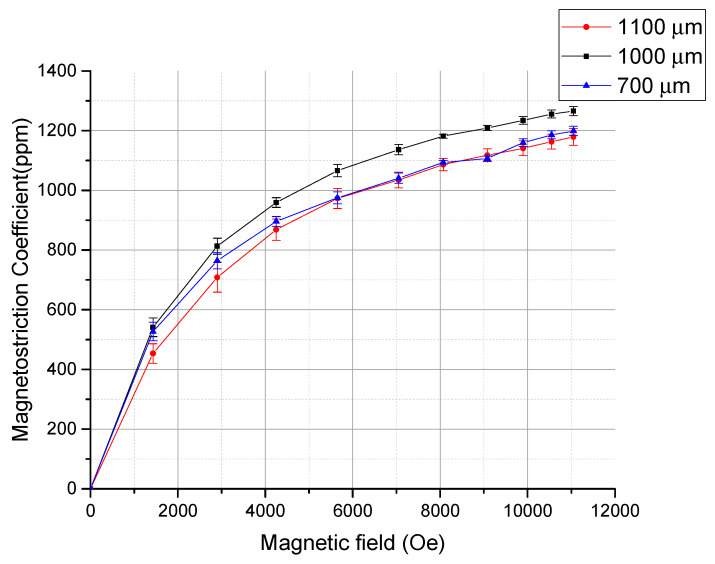
Magnetostriction coefficients of the Tb_0.36_Dy_0.64_Fe_1.9_ film obtained from three cantilevers with lengths 700, 1000, 1100 µm.

**Figure 11 micromachines-11-00523-f011:**
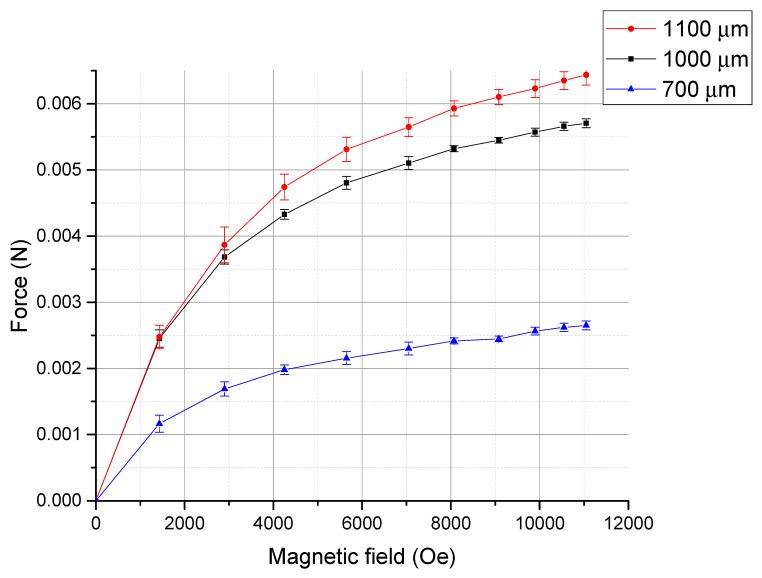
Generated forces of Tb_0.36_Dy_0.64_Fe_1.9_ obtained from three cantilevers with lengths 700, 1000, 1100 µm.

**Figure 12 micromachines-11-00523-f012:**
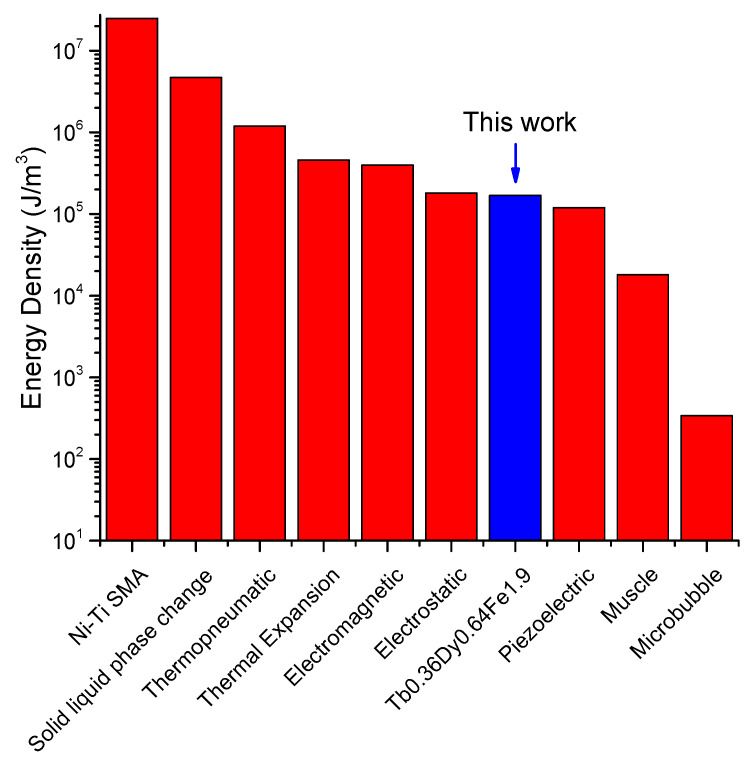
Comparison of energy densities for Tb_0.36_Dy_0.64_Fe_1.9_ and another types of actuators [40].

**Table 1 micromachines-11-00523-t001:** Typical dimensions of the fabricated cantilevered structure.

Cantilever Length	500 µm to 1.1 mm; 100 µm Step
Cantilever width	100 µm
SOI wafer dimension (Si/SiO_2_/Si)	1.5 µm/3 µm/550 µm
Cu Seed layer thickness	300 nm
Tb_x_Dy_(1−x)_Fe_y_ thickness	250 nm

**Table 2 micromachines-11-00523-t002:** Energy density of bulk Terfenol-D and electrochemical deposited Tb_0.36_Dy_0.64_Fe_1.9._

Materials	Power Density
Bulk Terfenol-D	5000 to 25,000 J/m^3^
Tb_0.36_Dy_0.64_Fe_1.9_ (700 µm-long cantilever at 11 kOe)	129,000 J/m^3^
Tb_0.36_Dy_0.64_Fe_1.9_ (1000 µm-long cantilever at 11 kOe)	169,000 J/m^3^
Tb_0.36_Dy_0.64_Fe_1.9_ (1100 µm-long cantilever at 11 kOe)	100,000 J/m^3^

**Table 3 micromachines-11-00523-t003:** Comparison of the magnetostrictive coefficients among bulk values, sputtered, and electrochemical deposited films.

Materials	Magnetostriction Coefficient (ppm)	Refs
Bulk Terfenol-D	1400	[1,2,7]
Electrodeposited Tb_0.36_Dy_0.64_Fe_1.9_ at 11 kOe	1250	This work
Sputtered Terfenol-D at 6 kOe	450	[24]
Sputtered Terfenol-D at 10 kOe	540	[23]
Sputtered Terfenol-D annealed 400 °C at 740 emu/cc	910	[20]
Sputtered Terfenol-D annealed 450 °C at 700 emu/cc	880	[19]
Electrodeposited Galfenol at 628 Oe	96	[6]
Bulk Galfenol	320~400	[5,6]
Sputtered Co_0.66_Fe_0.34_ annealed 800 °C	260	[11]
Electrodeposited Co_0.65_Fe_0.35_	1.5	[9]
Bulk TbFe_2_	2630	[1,2]

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
