# Peer review of "Magnetostrictive Performance of Electrodeposited TbxDy(1−x)Fey Thin Film with Microcantilever Structures"

_micromachines, 2020, doi:10.3390/mi11050523_

Round 1

Reviewer 1 Report

The authors present a work on the magnetostriction properties of self standing microcantilevers fabricated with electrodeposited terfenol-D. The general planning of the experiment is adequate. The methods are also adequate, although the description of the measurements and the results representation can be improved in order to further support the conclusions of the work. There are also flaws under my point of view related with the data analysis performed. Overall, I think that the manuscript is interesting and after major revision it would be suitable to be published in Micromachines.

Here I introduce more specific comments:

1) In the experimental section, it would be useful for the community to know the specific commercial electrolyte used by the authors in order to deposit the Terfenol-D layer. (Lines 62-63)

2) It would be useful to compare the saturation magnetization measured with the bulk terfenol-D one in order to further corroborate the good results of the electrodeposition optimization process.

3) Also related with the hysteresis loop measurement, it would be good to indicate the measurement direction (i.e. magnetization measured with the field applied in-plane / out-of-plane). Related with this, I assume that the curve shown in figure 3 corresponds to an in-plane measurement. If this is the case, have the authors measured the response in the orthogonal direction? Does exhibit the system any in-plane /out-of-plane anisotropy that could be important in order to analyse different magnetostriction-active regimes depending on the sample orientation?

4) Related with this, I assume that the general magnetic characterization measurements where performed in a continuous control film. Have the authors checked that the properties of the samples after all the fabrication process are similar to the continuous film ones? Also, it is important to be sure that the magnetic configuration present in the microstructured cantilevers is not modifying too much the magnetic behaviour of the continuous film. This also could affect the range where the magnetostrictive effects manifest most.

5) The sentence beginning in line 142 is very confuse and needs to be clarified. The specific sentence is "The magnetic field was applied to left to rigth and parallel direction of cantilever."

6) Have the authors performed the magnetostrictive measurements with the field being applied both parallel and perpendicular to the longitudinal axis of the cantilevers? This is related with he previous point and it should be clarified, as authors indicate that they are considering that the deflection in the perpendicular direction to the cantilever axis is negligible. Is this an assumption or it comes from an observation?

7) In figure 9 the vertical displacement of the cantilever (unbending) is presented as function of the external magnetic field. The authors state that the effect saturates around 7900 Oe in agreement with the saturation shown by the control film. However, above that level, the magnetostriction is not yet saturated as the authors state. In fact, I would like to ask specifically about the behaviour shown by the 1000 um cantilever, especially for large fields. The measurement shows an increase in the vertical deflection out of the general trend. Is this an error or it is real? If it is real, what is happening with the system?

8) Could be the magnetostriction coefficient overestimated due to the assumption of a negligible perpendicular deformation? How is this assumption (if it is an assumption, check comment number 6) affecting the results?

9) Is it correct to assume a beta factor for the neutral plane of the cantilever of 1/2 when the cantilever is formed by Si and Terfenol-D? The authors indicate that Terfenol D has a Young’s modulus 3 times larger than the Si one. On this regard, have the authors considered to measure the intrinsic mechanical properties of the cantilever's materials (especially the electrodeposited Terfenol-D which is the main actor in this work) to increase the accuracy of the indirectly identified properties?

10) Could the authors explain in detail why the behaviour indicated in table 2? It is simply mentioned that the Energy density which could be stored in the 1000 um is 220000 J/m^3. Is this related with the final trend shown by this cantilever in figure 9? This is an important point and it must be completely clarified.

Reviewer 2 Report

The presented study “Magnetostrictive Performance of Electrodeposited TbxDy(1-X)Fey Thin Film with Microcantilever Structures” by Shim et al. investigates the growth and performance of a electroplated TbxDy(1-X)Fey film on a Cr-Cu seed layer.

The study is well presented and lays out a new manufacturing routine for a high-stress thin film system, whose magnetostriction is comparable to or higher than other commonly used magnetostrictive materials. I generally support publication in Micromachines, however, there are a few comments I would like to see addressed before accepting the article:

1) The mentioned film thicknesses throughout the paper vary between 200, 250, 300 nm. Please clarify if the analysis involves different thicknesses and why not only one consistent thickness was used.

2) Please add error bars or a comment why they are not required to Fig. 2, 9, 10, and 11.

3) Line 95: The mentioned saturation field seems a little high looking at Fig. 3. How was this determined?

4) There is something wrong in the caption of Fig. 4.

5) Line 120-122: Those are bulk values if I understand correctly. Should those not differ in case of thin films of only a few hundred nm thickness? Please clarify and add a discussion if necessary.

6) General comment on the figures: The captions should be more self-explanatory. Currently, they are a little minimalistic. Especially Fig. 6 also uses abbreviations that are not introduced in the text. They are easy to guess, but should still be introduced in the caption.

7) Line 165: The 1300 ppm was presumably not determined at the saturation field that was shown in Fig. 3 but rather at the highest field.

8) Table 2: Why is the value for the 1000 um cantilever so much higher? Please clarify and add a statement to the paper.

9) Fig. 12: There seem to be some text/arrows misplaced.

10) Line 192: That seems a bit arbitrary. What PZT are the authors using as reference?

11) Please double-check the references. E.g. ref. 36 should be Meníc, not Menc.

12) Please do a thorough proof-reading of the text and send it through a spell checker. There are still many typos and grammar issues. Also, please correct the subscript in the title.

Reviewer 3 Report

referee report
micromachines-773353
Hang Shim et al.
Magnetostrictive performance of electrodeposited...

The present manuscript describes the fabrication of electrodeposited, magnetostrictive Terfenol-D actuators. This is an interesting topic, well suited for Micromachines. Overall, the manuscript looks to be well organized and well prepared. The English, however, requires attention, and a check of the manuscript should be done by a native speaker to avoid all the obvious grammatical problems -- it already starts in the abstract!

The manuscript would also have required a more proper re-reading before submission -- this would have avoided such silly mistakes like in the title.
Now, to the details which appear when carefully reading the manuscript:

- The authors mistook the experimental section for sample preparation only. In this section, there should be a complete description of all the apparatuses employed in the research. There are no details given concerning microscopy (optical and AFM), magnetometry, etc.
- An EDX-diagram should be properly replotted; a simple screenshot is unacceptable for publication.
- There is a proper sign for micrometer -- µm. This should be used in all diagrams and in the main text.
- Figure 4. Please check the caption. Anyway, it should be more descriptive. How did you analyse the grain size mentioned. For sure, it is the mean grain size. There should be a proper diagram of this analysis with a proper
fit to see the full distribution.
- Figure 5: A MFM-image is by no means useful to give information about the grain size. It might be possible to discuss grain coupling etc., but there is absolutely no information about the conditions the image is taken in, nor
there is a discussion what one can see in this image. Without a proper description (we even do not know the type of cantilever employed), this image is fully useless.
- There should be always a space between physical quantity and its unit. Furthermore, the units are never written in italics.
- Fig. 7. The SEM images are not sharp, of low contrast, and have scale bars which are difficult to be read.
- Figs. 10 and 11 do not have units on the x-axes.
- the abbreviation of equation is Eq. or Eqs.
- the reference list is not formatted according to the rules of the journal. Furthermore, the entries are not treated consequently in the same style: author prenames, journal names, etc.

Overall, the manuscript contains interesting data which should be published, but the technical parameters of the manuscript do not allow publication in the present form.

Round 2

Reviewer 1 Report

The authors have improved the manuscript but I think that some important points indicated in the previous review round have not been properly addressed. Due to this I recommend a second revision round in order to further improve the quality of the work.

1) First of all, in order to allow experiment reproducibility it is crucial to clearly indicate the electrolyte solution used as this is directly related with the final properties of the fabricated system. Authors indicate in the manuscript that more information can be found in reference 31, but this one is a conference contribution which, at least I have not been able to access. If the information is not accesible in a previous work or in the literature, authors should indicate the accurate concentrations of each specific chemical agent used.

2) I have been checking for the values of Saturation Magnetisation of bulk Terfenol-D in the manuscript at the specific position indicated by the authors in the response letter, but I have not found it there or anywhere.

3) About the response to comment 3, the authors clearly state now the orientation of the VSM measurement but they are not commenting anything about the magnetic anisotropy point. Is expected in this material to exist any specific magnetic anisotropy that potentially could affect to the magnetostrictive properties? Authors should support their response either with measurements or previous experimental evidences form them or other groups.

4) Finally, if the unexpected behaviour from the 1000 um cantilever comes from cantilever lenght errors due to the fabrication process, could the authors measure the total cantilever length as it is after the fabrication process in order to take into account the real size of the sample? This could also affect the other cantilevers if the error is associated with the fabrication process, and it is something which seems to be relatively simple to be done by using for instance the optical microscope used to measure the deflection of the cantilever.

Reviewer 2 Report

Most of my comments have been answered. However, two of them remain:

1) page 6, line 133ff: OK, I understand that your films are already thick enough and also that the thin film value would be difficult to determine accurately. Since, however, there are cases in the literature (e.g. doi:10.1088/1742-6596/214/1/012049) that report a change in the Young's modulus with thicknesses <100nm, I suggest to alter the line (page 6, line 133ff) so that it states that the Young's modulus is expected to be already similar to the bulk value in your case due to the thickness not falling into that ultra-thin film limit.

2) There are still extensive English language corrections necessary. Sorry, but it does not look like you did a real grammar check on the text - and neither in your rebuttal letter.

And one more comment:

3) Page 12, line 226 has not been adapted with the main text value of 1250ppm.

Reviewer 3 Report

The authors have dealt well with the comments raised by the referee, and the manuscript was modified accordingly. This is positive, but there are still many technical deficits which need to be fixed prior to publication:

1.) There should always be a space between a physical quantity and its unit.

2.) The references should be numbered in a proper order -- currently this is a mess.

3.) The reference list is a pure catastrophe.

The journal style is not followed, no proper journal abbreviations were used, capital letters are used, and even worse, several references are copied from other works without removing former subscripts like "a", "b", etc. (see Ref. 5).

Do the authors really know how to prepare a manuscript?

Round 3

Reviewer 1 Report

The authors have improved the manuscript. However, I would like to point some issues that could be corrected, as well as very important point regarding the characterisation procedure for the main result of the paper. Due to this I recommend still a revision of the paper.

First, when the authors state in their response the magnetic anisotropy of the system, they are talking about the shape anisotropy due to the magnetic film configuration. Authors also mentioned that they have a hard plane in the perpendicular direction regarding to the film's plane and an easy direction within the plane of the sample. If the only responsible anisotropy is the shape anisotropy, then this explanation is wrong as the easy plane is the sample's plane and the hard direction is the perpendicular one. However, I have found some previous works in the literature that indicate important effects due to magnetocrystalline anisotropy related with magnetoelastic phenomena in terfenol-D (for instance Appl. Phys. Lett. 98, 012503 (2011)). Maybe it would be good to indicate that the authors are not taking into account any magnetocrystalline anisotropy effect in their calculations although these could be important.

Other technical point, authors present the saturation magnetisation of bulk terfenol-D from the literature in Teslas while their magnetisation curve is presented in Gauss. For convenience both magnitudes should be presented in the same units. Other point related with this is an explanation of why a difference of around a 40% in the Ms is present, when the authors are reporting excellent magnetistriction coefficient values. How both quantities scale? Are the obtained results sensible? An explanation is needed here.

My main and most important concern is related with the characterisation of the cantilever's initial length, thus the one which is going to be scaling the magnetostriction coefficient measurement for each cantilever. If as the authors state, this initial characterisation has been done poorly, the associated error with the final magnetostriction could be very important. What is the error in the determination of the initial cantilever's length, so the error associated with the l parameter in equation 1? How is the error on this parameter affecting the final result? I want to emphasise this point, as it is associated with the main point of the work.
